# Low-level Arctic clouds: A blind zone in our knowledge of the radiation budget

Hannes J. Griesche[1], Carola Barrientos-Velasco[1], Hartwig Deneke[1], Anja Hünerbein[1], Patric Seifert[1], and Andreas Macke[1]

[1]Remote Sensing of Atmospheric Processes, Leibniz Institute for Tropospheric Research, Leipzig, Germany

**Correspondence:** Hannes J. Griesche [griesche@tropos.de]

**Abstract.** Quantifying the role of clouds in the Earth's radiation budget is essential for improving our understanding of the drivers and feedbacks of climate change. This holds in particular for the Arctic, the region currently undergoing the most rapid changes. This region, however, also poses significant challenges to remote-sensing retrievals of clouds and radiative fluxes, introducing large uncertainties in current climate data records. In particular, low-level stratiform clouds are common in the
5 Arctic but are, due to their low altitude, challenging to observe and characterize with remote-sensing techniques. The availability of reliable ground-based observations as reference is thus of high importance. In the present study, radiative transfer simulations based on state-of-the-art ground-based remote sensing of clouds are contrasted to surface radiative flux measurements to assess their ability to constrain the cloud radiative effect. Cloud radar, lidar, and microwave radiometer observations from the PS106 cruise in the Arctic marginal sea ice zone in summer 2017 were used to derive cloud micro- and macrophysical
properties by means of the instrument synergy approach of Cloudnet. Closure of surface radiative fluxes can only be achieved by a realistic representation of the low-level liquid-containing clouds in the radiative transfer simulations. The original, likely erroneous, representation of these low-level clouds in the radiative transfer simulations led to errors in the cloud radiative effect of $54\,\mathrm{W\,m^{-2}}$. In total, the proposed method could be applied to 11% of the observations. For the data, where the proposed method was utilized, the average relative error decreased from 109% to 37% for the simulated solar and from 18% to 2.5%
for the simulated terrestrial downward radiative fluxes at the surface. The present study highlights the importance of jointly improving retrievals for low-level liquid-containing clouds which are frequently encountered in the high Arctic, together with observational capabilities both in terms of cloud remote sensing and radiative flux observations. Concrete suggestions for achieving these goals are provided.

## 1 Introduction

In the past 30 years, the surface temperature in the Arctic has increased by more than twice the globally averaged increase. In addition, the differential temperature rise has intensified over the same period (Chylek et al., 2022). This phenomenon of the increased warming in the Arctic is known as Arctic amplification and is attributed to several feedback mechanisms (Wendisch et al., 2017; Goosse et al., 2018). Clouds play a complex role in the context of Arctic amplification. On the one hand, clouds

influence other processes and feedback mechanisms driving the rapid changes in the Arctic, such as the ice-albedo feedback
(He et al., 2019; Kay et al., 2016). On the other hand, clouds directly impact the atmospheric radiative fluxes.

One measure of the impact of clouds on the radiation budget is the cloud radiative effect (CRE). The macro- and microphysical properties of clouds (e.g., phase, particle shape and size, vertical extent) strongly influence the magnitude of the CRE, as well as the interaction of radiative fluxes with surface properties (e.g., surface albedo, skin temperature). A way to determine the CRE is to measure the radiative fluxes below and above the cloud, as it can be done, for example, with tethered balloon
platforms (Egerer et al., 2019; Lonardi et al., 2022) or aircraft (Becker et al., 2023). The CRE can the be determined by comparing the different profiles. Here, the temporal difference between the measurements due to the ascent or descent time of the platform needs to be considered. A more common approach to study the CRE is the utilization of radiative transfer simulations (e.g., Kay and L'Ecuyer, 2013; Shupe et al., 2015; Ebell et al., 2020; Barrientos-Velasco et al., 2022). Such simulations are based on the input of cloud properties. The simulations provide vertically resolved radiative fluxes for the same period for both
cloudy and cloud-free conditions. The simulated fluxes can be evaluated, e.g., against surface measurements. By contrasting the cloudy and cloud-free scenarios, the radiative effect of clouds can be determined.

Satellite products of cloud properties and radiative fluxes are available for the entire Arctic. Active satellite retrievals based on cloud radar and lidar synergies, such as DARDAR (Cazenave et al., 2019) or CAPTIVATE (Mason et al., 2023), can retrieve the vertical structure of cloud microphysical properties. The applied cloud radar, however, can suffer from ground
clutter and the lidar can lose sensitivity close to the ground, which induces challenges to resolve low-level clouds (Liu et al., 2017). Additionally, approaches for passive sensors are established (e.g., Kato et al., 2018; Stengel et al., 2020), yet, passive sensors can have difficulties to resolve the vertical structure of the clouds (Yost et al., 2021). Vinjamuri et al. (2023) compared cloud properties from passive satellite observations and ground-based remote sensing at four Arctic sites. The authors showed an agreement of the cloud fraction of clouds with an optical thickness of 3 or higher of better than 90%. In addition, they
highlighted that the differences in the derived cloud top heights are generally less than 500 m. Based on 34 years of satellite observations, Philipp et al. (2020) found an increasing trend of Arctic low-level clouds located below 680 hPa, which induced a warming trend at the surface. Kay and L'Ecuyer (2013) applied a combination of active and passive satellite observations and found an annual-mean surface net warming effect of clouds over the Arctic Ocean between 2000 and 2011 by 10 W m$^{-2}$. Lelli et al. (2023) assessed the CRE of Arctic clouds based 20 years of satellite observations. The authors observed a trend of more
liquid clouds over the open ocean, inducing a cooling effect at the surface. The authors pointed out that this effect has seasonal and regional differences and that the effect is weaker above closed ice areas and the marginal ice zone and strongest in summer. Yet, investigations of small-scale processes require the application of models and measurements with a smaller footprint, as ground-based remote-sensing approaches offer. Shupe et al. (2015) and Ebell et al. (2020), for instance, each have investigated 2 years of ground-based remote sensing and radiative transfer simulations at the land-based sites in Utqiaġvik, USA, and Ny-
Ålesund, Svalbard, respectively. Barrientos-Velasco et al. (2022) studied radiative fluxes observed during the Polarstern cruise PS106 (cruise track is shown in Fig. 1, Wendisch et al., 2019) performed in May - July 2017 in the marginal sea-ice zone, north and north-east of Svalbard, of the Arctic Ocean and contrasted them to radiative transfer simulations as well as satellite observations. The surface flux differences reported in these studies, averaged over the investigated period, between simulations

and observations were within a range of $\pm 23\,\mathrm{W\,m^{-2}}$ for the solar and $\pm 7\,\mathrm{W\,m^{-2}}$ for the terrestrial radiative fluxes. Shupe et al. (2015) reported that the largest biases were found for clear-sky and ice-cloud situations. In Ebell et al. (2020) a large difference between the observed and simulated fluxes were found during the summer months, which was attributed to clouds missed by the observations. Barrientos-Velasco et al. (2022) reported similar challenges for the ground-based observations for low-level mixed-phase clouds and ice clouds. Huang et al. (2022) compared radiative fluxes derived form satellites observations to those measured on the ground, during the year-long Multidisciplinary drifting Observatory for the Study of Arctic Climate (MOSAiC) expedition (Shupe et al., 2022). The authors reported an average surface flux difference between the satellite-based and ground-based retrievals of $\pm 15\,\mathrm{W\,m^{-2}}$ for April to September 2020. Differences in the upwelling radiative fluxes were partially attributed to an underestimated surface albedo in the satellite footprint and differences in the downwelling fluxes to an underestimation of the atmospheric optical thickness in the satellite retrieval.

The dominant contribution to the Arctic surface CRE is caused by low-level mixed-phase clouds (Shupe and Intrieri, 2004). In modeling studies, it has been shown that these clouds can provide a critical contribution to extreme melting events of the Greenland ice sheet (Bennartz et al., 2013) and have increased the surface downward terrestrial radiative fluxes during this event by $100\,\mathrm{W\,m^{-2}}$ (Solomon et al., 2017). Additionally, Turner et al. (2007) showed the necessity of an accurate representation of low-level liquid-containing clouds with a liquid-water path (LWP) below $0.1\,\mathrm{kg\,m^{-2}}$ in radiative transfer studies. The authors used remote sensing and models to highlight the sensitivity of the radiative effect of these clouds to small LWP perturbations and the challenge of accurately deriving the cloud microphysical properties. The properties of low-level mixed-phase clouds are subject to boundary-layer processes and the radiative forcing produced by higher-level clouds above (Griesche et al., 2021; Shupe et al., 2013; Yu et al., 2019), and their presence is critical to atmospheric stability (Sedlar, 2014), surface conditions (Solomon et al., 2017), as well as large-scale processes (Huang et al., 2021).

Only a few ship-based studies have been performed in the Arctic Ocean with the ability to continuously derive height-resolved cloud microphysical properties, i.e., were equipped with a collocated cloud radar and lidar. Low-level clouds in the Arctic have been observed during the aircraft campaign ACLOUD (Arctic CLoud Observations Using airborne measurements during polar Day, Wendisch et al., 2019), which was performed simultaneously with the first month of the PS106 cruise (Mech et al., 2019). Even though, also limited to clouds above $150\,\mathrm{m}$, during ACLOUD a peak of low-level clouds just above the lowest detection range of the applied cloud radar was observed. Shupe et al. (2005) reported for the year-long Arctic ice drift SHEBA (Surface Heat Budget of the Arctic Ocean, Uttal et al., 2002) performed in 1997 and 1998 that the lowest detectable cloud base was at $105\,\mathrm{m}$. For similar, but shorter campaigns, as the Arctic Summer Cloud Ocean Study (ASCOS, Tjernström et al., 2014) and the Arctic Ocean expedition (AO2018, Vüllers et al., 2021) performed in 2008 and 2018, respectively, the lowest detected range gate was around $150\,\mathrm{m}$, with no height-resolved microphysical properties derived for clouds below that height (Shupe et al., 2013; Vüllers et al., 2021). During the ACSE (Arctic Clouds in Summer Experiment, Tjernström et al., 2015) campaign conducted in 2014, a cloud radar was operated, which had its lowest range gate at $80\,\mathrm{m}$ and a maximum height of $5980\,\mathrm{m}$ (Achtert et al., 2020). Based on the measurements taken during this campaign Cloudnet had identified an unusually high frequency of aerosol and insect occurrence (for the Arctic), despite the rather low cloud radar detection limit. This has been attributed to missing cloud identification (Achtert et al., 2020). To account for these miss-classifications the

occurrence of fog was identified by an in-situ visibility sensor on the ship. Yet, due to their very low altitude, these clouds still pose challenges to state-of-the-art remote-sensing approaches. An Arctic-wide quantification of these low-level clouds and the disentangling of their radiative effects from those from higher clouds is still difficult. By means of lidar observations performed during the PS106 cruise, an occurrence of clouds located below an altitude of 165 m, i.e., below the lowest detection range of most remote-sensing techniques, during 25% of the observational time was determined (Griesche et al., 2020). Griesche et al. (2020) elaborated that these low-level clouds with occurrence heights between around 20 and 150 m above ground are located in the blind zones of many ground-based, space-borne, and airborne remote-sensing techniques. Hence, their spatial extent was to date not quantifiable. It is likely that they cover large portions of the marginal sea ice zone where humid marine air masses pass over the cold sea ice.

In this manuscript, we will demonstrate the relevance of the low-level clouds on the CRE by means of a selected case study. We propose a method to reduce downward radiative flux biases of low-level stratus clouds (LLS) by evaluating the flux differences between 1-D radiative transfer simulations and observations collected during the PS106 cruise. The simulations were performed with the TROPOS (Leibniz Institute of Tropospheric Research) Cloud and Aerosol Radiative effect Simulator (T-CARS) (Barlakas et al., 2020; Witthuhn et al., 2021; Barrientos-Velasco et al., 2022). Cloud properties derived by the instrument synergy approach Cloudnet (Illingworth et al., 2007; Tukiainen et al., 2020) served as realistic input parameters for the radiative transfer simulations and the surface radiation measurements of the OCEANET-Atmosphere facility (hereafter referred to as OCEANET) as true validation data. Cloudnet combines active and passive remote-sensing observations to derive macro- and microphysical cloud properties. To address the challenges of Arctic clouds, especially the frequent occurrence of optically-thick, low-level clouds, the standard Cloudnet output had to be adjusted. Therefore, new approaches to derive the ice-crystal effective radius ($r_{\mathrm{eff,ice}}$) and for the detection of LLS were introduced in Griesche et al. (2020), added to the Cloudnet processing chain and published via the long-term archive Pangaea Griesche et al. (2020b, f). While using Cloudnet products in the radiative transfer model led to cases with good agreement between simulated and observed radiative fluxes at the surface during the PS106 cruise, there were other cases where the biases were larger than the radiometer instrumental uncertainties. Here, we quantify the contribution of low-level liquid-containing clouds to the observed differences between simulated and observed surface radiative fluxes. Therefore, an effective improvement of the Cloudnet cloud properties was used to simulate radiative fluxes during the PS106 cruise by applying the additional information on LLS clouds presented in Griesche et al. (2020) to T-CARS. These model results are compared to a control simulation without the improved low-level cloud treatment. This approach allows us to determine the surface CRE caused by low-level mixed-phase clouds.

Section 2 gives an introduction to the applied observations, the radiative transfer simulations, and the treatment of the low-level stratus clouds. In Sect. 3 the resulting surface CRE is presented by means of a case study obtained during the PS106 cruise. First, the detection of liquid clouds and the quantification of their properties is introduced. In the next step, their relevance for the radiative transfer simulations is evaluated. Additionally, an overview of the applicability and the resulting effect on the radiative transfer simulation of the presented method for the whole PS106 campaign is provided. A discussion of the results and general conclusions are given in Sect. 4.

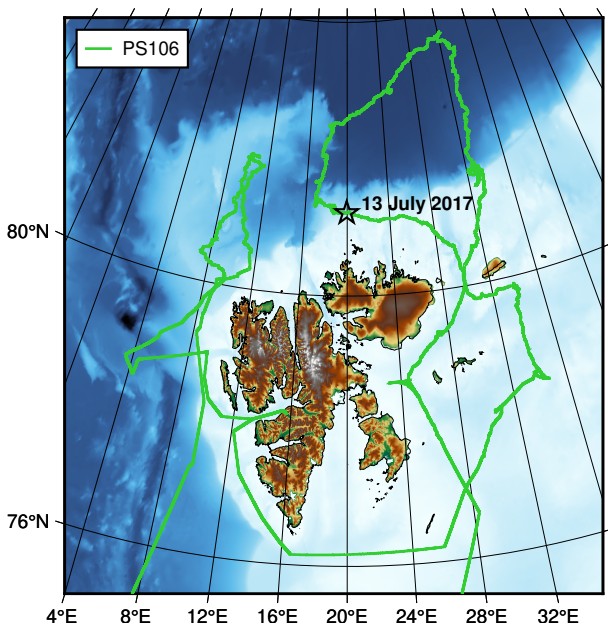

**Figure 1.** Cruise track of the PS106 expedition between 29 May 2017 and 18 July 2017. The black star marks the position during the presented case study on 13 July 2017. Map created with PyGMT (Tian et al., 2023)

## 2 Methods

### 2.1 OCEANET observations

In this study, the required cloud properties were derived based on the instrument synergy approach of Cloudnet. During the PS106 cruise, the mobile remote-sensing supersite OCEANET from TROPOS performed continuous observations of the atmospheric structure (Griesche et al., 2020). By default, OCEANET is equipped with a multiwavelength Raman lidar Polly$^{\mathrm{XT}}$ (Engelmann et al., 2016), a microwave radiometer HATPRO (Rose et al., 2005), and broadband pyranometer and pyrgeometer. For the PS106 cruise, OCEANET was complemented for the first time with a motion-stabilized and heave-corrected cloud radar

MIRA-35 (Görsdorf et al., 2015; Griesche et al., 2020). This data was processed by Cloudnet in order to derive cloud macro- and microphysical properties, such as the liquid-water content (LWC) and the ice-water content (IWC), and the liquid-droplet effective radius ($r_{\mathrm{eff,liq}}$), and $r_{\mathrm{eff,ice}}$. The liquid and ice cloud microphysical properties were derived based on the combination of cloud radar, lidar, microwave radiometer, and radiosonde observations. The IWC and $r_{\mathrm{eff,ice}}$ are based on an empirical relationship between the radar reflectivity factor and temperature (Hogan et al., 2006; Griesche et al., 2020). The LWC is retrieved

by scaling the MWR LWP adiabatically on the liquid-containing cloud. The $r_{\mathrm{eff,liq}}$ is retrieved using the cloud radar reflectivity and the assumption of a marine stratus cloud with a cloud droplet number concentration of $100\,\mathrm{cm}^{-3}$ (Frisch et al., 2002).

## 2.2 T-CARS simulations

T-CARS is a Python-based environment for simulating vertically resolved broadband radiative fluxes and heating rates for cloudy and cloud-free conditions from the surface to the top of the atmosphere. The radiative transfer simulations were performed by means of the 1-D Rapid Radiative Transfer Model for General Circulation Model applications (RRTMG, Barker et al., 2003; Clough et al., 2005; Mlawer et al., 1997) which has been implemented into T-CARS. For this study, near-surface temperature and pressure measured aboard Polarstern, extrapolated atmospheric properties of humidity, temperature, and pressure from the radiosondes launched every 6 hours throughout the whole cruise from Polarstern, and atmospheric trace-gases profiles (i.e., Anderson et al., 1986) were used as input to T-CARS. The input parameter for the surface albedo is based on the collocated data to the ship location of CERES Synoptic 1-degree Ed. 4.1 products (Minnis et al., 2021). Additionally, cloud properties like LWC, IWC, $r_{eff,liq}$, and $r_{eff,ice}$ are necessary to perform the simulations. Here, the cloud properties derived by Cloudnet, based on the remote-sensing observations were applied. The method implemented in this analysis first compares the simulated radiative fluxes with observed values of downward solar (SD) and terrestrial (TD) and then derives the CRE at the surface, following Barrientos-Velasco et al. (2022). The current study defines the CRE as the difference between an all-sky and a clear-sky atmosphere.

## 2.3 Improved low-level stratus liquid microphysical properties for radiative transfer simulations

For the realization of reliable radiative transfer simulations, an accurate representation of the atmospheric state in the model is necessary. The nature of Arctic clouds, especially the optically thick, low-altitude clouds, poses challenges on the task to derive the cloud microphysical properties for the entire tropospheric column. Strong lidar signal attenuation inside the LLS makes the continuous application of existing $r_{eff,ice}$ retrievals which apply lidar-radar instrument synergy, as used, e.g., for the DARDAR-CLOUD algorithm (Cazenave et al., 2019), impossible. Hence, $r_{eff,ice}$ was derived based on cloud radar measurements alone, as proposed by Griesche et al. (2020). This method ensures the continuous identification of microphysical properties up to cloud top. The low altitude of the clouds, which was frequently below the lowest range gate of the cloud radar, was addressed using the near-range capabilities of the lidar Polly$^{XT}$. The near-range channel allowed a cloud detection down to a height of 50 m above the instrument and to adjust the cloud base height accordingly (Griesche et al., 2020). This approach is applied to the whole campaign and analyzed in detailed for a case study on 13 July 2020. The location of Polarstern during the case study is marked by the black star in Fig. 1 and the observations are shown in Fig. 2.

The liquid phase detection in Cloudnet is based on the observed lidar attenuated backscatter coefficient, but the retrievals for LWC and $r_{eff,liq}$ rely on the cloud radar reflectivity. The lowest height range of the cloud radar, however, is located 165 m above the ground. In addition, in the case of a complete lidar signal attenuation below the lowest cloud radar range gate, no liquid phase is identified by Cloudnet in the whole column, as it is the case in Fig. 2 around 05:00 UTC and often between 07:50 – 09:30 UTC. Consequently, no liquid-water cloud microphysical properties were derived. Therefore, the LLS cloud mask was used to identify the presence of a liquid-water cloud below the lowest range gate of the cloud radar. In the case of a detected LLS, the column integrated LWC (hereafter denoted as LWP$_{LWC}$) was compared to the LWP derived by the MWR

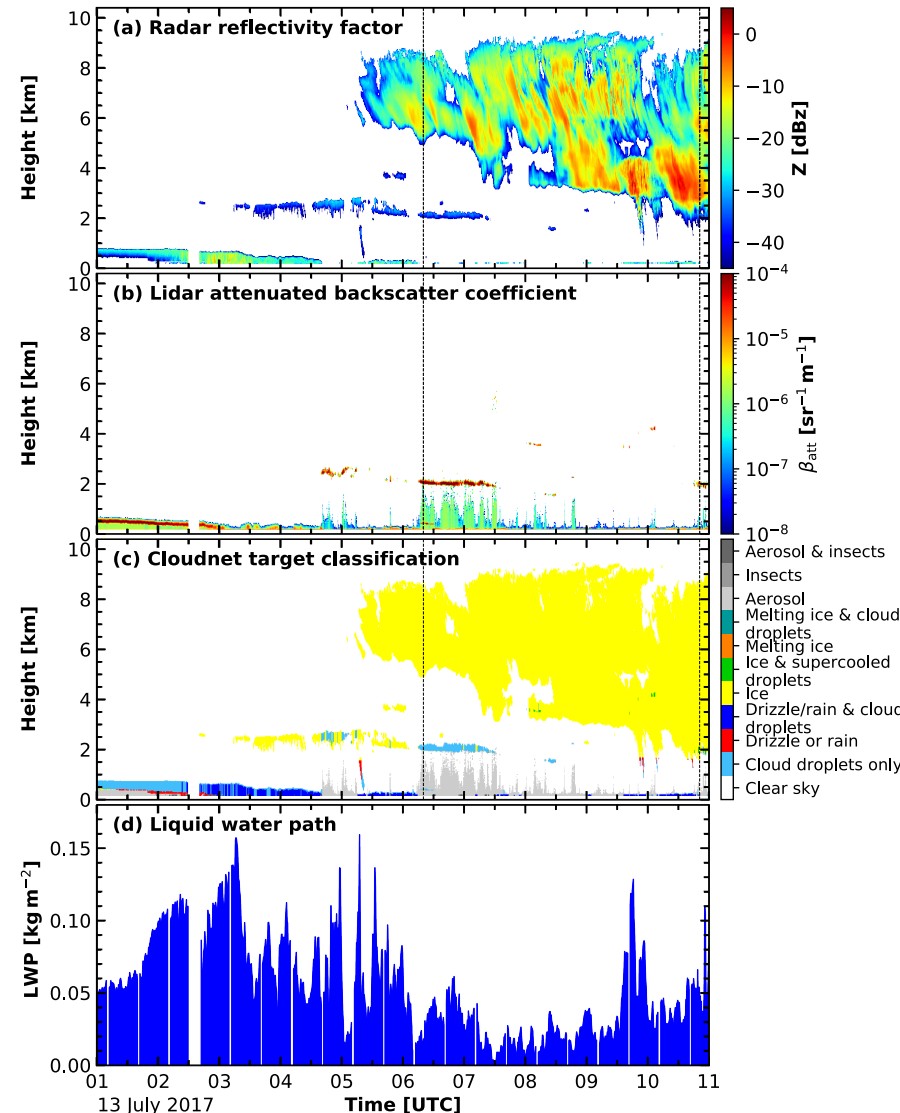

**Figure 2.** Profiles of cloud radar reflectivity (a), lidar attenuated backscatter coefficient (b), and Cloudnet target classification (c) between 0 and 10 km height, and LWP (d) for 13 July 2017 01:00 UTC to 11:00 UTC. The two dashed lines mark the time of the radiosonde launches for the profiles shown in Fig. 4 (b) and (c).

HATPRO. The LWC derived by Cloudnet is scaled to the LWP from HATPRO. Hence, both quantities are identical if Cloudnet had identified a liquid-water cloud. If no liquid-water cloud was identified, no LWC was derived by Cloudnet, i.e., $LWP_{LWC}$ is equal to zero. Yet, such a strong lidar signal attenuation can only be caused by the presence of a liquid-dominated cloud layer. In this case, $r_{eff,liq}$ was estimated using the difference between the LWP from HATPRO and $LWP_{LWC}$ (denoted as $\Delta LWP$).

To determine a representative LWP-$r_{\text{eff,liq}}$ relationship, the $r_{\text{eff,liq}}$ product as derived by Cloudnet for surface-coupled low-level stratus clouds during the PS106 cruise was analyzed. Surface-coupled clouds were defined as clouds with a quasi-constant potential temperature $\theta$ profile below the liquid-dominated cloud layer, following Gierens et al. (2020). The potential temperature profile was calculated from the temporal closest radiosonde, which were launched every 6 hours. A surface-coupled cloud was identified if the difference between the cumulative mean of $\theta$ and $\theta$ did not exceed $0.5\,\text{K}$ between the surface and the liquid-dominated cloud base. In addition, only clouds with a liquid-dominated layer less than $200\,\text{m}$ thick were analyzed. In Fig. 3 the resulting distribution of $r_{\text{eff,liq}}$ for LWP between 0 and $0.3\,\text{kg\,m}^{-2}$ is shown. Based on these results, a linear LWP-$r_{\text{eff,liq}}$ relationship for $\Delta$LWP below $0.15\,\text{kg\,m}^{-2}$ was applied and values between 5 and $15\,\mu\text{m}$ were used for $r_{\text{eff,liq}}$. For $\Delta$LWP larger than $0.15\,\text{kg\,m}^{-2}$ a constant $r_{\text{eff,liq}}$ of $15\,\mu\text{m}$ was utilized. Finally, the liquid cloud microphysical properties for the radiative transfer simulations were estimated as follows. For each time step a height-constant $r_{\text{eff,liq}}$ derived by the LWP-$r_{\text{eff,liq}}$ relationship was applied to the layer determined by the LLS identification and the LWC was determined by an adiabatic scaling of $\Delta$LWP inside the LLS boundaries.

## 3 Results

### 3.1 Case study: 13 July 2017 - Signal attenuation by LLS

To evaluate the improved procedure for improved CRE retrieval, and to highlight the benefits of the introduced low-level mixed-phase cloud detection and the estimation of the microphysical properties, its application is presented here for a case study from 13 July 2017. In Fig. 2 an overview of the cloud observations between 01:00 UTC and 11:00 UTC is presented. The

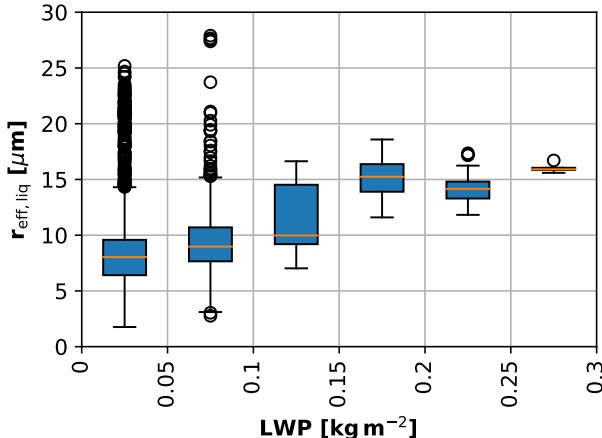

**Figure 3.** Box plot depicting the $r_{\text{eff,liq}}$ distribution from optically thick surface-coupled low-level clouds during the PS106 cruise, for different LWP. The median value is represented by the orange bar. The blue boxes show the 25 and 75 percentile and the caps mark the minimum and maximum values (circles show outliers).

corresponding radiosonde profiles up to 4 km height are given in Fig. 4. Figure 5 shows the derived Cloudnet liquid and ice microphysical products.

The period started with a liquid-water-dominated stratus cloud layer, which was located between 0.5 km and 1 km height and slowly descended towards lower altitudes. At the beginning of this period, the layer was thermodynamically decoupled from the surface, as can be seen by the $\theta$ profile in Fig. 4 (a). With decreasing cloud base height, this layer became coupled to the surface (Fig. 4 (b)), and precipitation formed after 02:00 UTC. Between 04:30 UTC and 05:30 UTC and after 06:30 UTC the entire cloud layer was below the lowest detection limit of the cloud radar, and hence of Cloudnet, and was therefore not identified by the original classification. Above this layer, at 2.5 km height and between 03:00 UTC and 08:00 UTC an altocumulus cloud was observed. This cloud was only classified as 'mixed-phase' (green) or 'liquid' (blue) when the lidar was able to penetrate this layer. In the case of complete lidar signal attenuation in the layer below, the altocumulus layer was classified as 'ice' cloud (yellow). The missing liquid-water identification is reflected in the Cloudnet products as presented in Fig. 5. After 04:30 UTC, LWC (Fig. 5 (a)) and $r_{\mathrm{eff,liq}}$ (Fig. 5 (b)) of the altocumulus layer as well as of the LLS cloud deck was only occasionally determined.

In Fig. 6 (a) the simplified Cloudnet classification mask (above 165 m) combined with the LLS cloud classification mask (below 165 m) is shown. This mask revealed the presence of an LLS cloud almost continuously during the entire period after 02:00 UTC. Only during a short situation of very few or no low clouds from 06:30 UTC to 07:30 UTC no LLS was identified. Figure 6 (b) depicts the LWP derived by HATPRO in blue and the difference $\Delta$LWP which is shown in orange. Two periods with increased $\Delta$LWP can be identified. The first period was observed between 04:30 UTC and 05:30 UTC with $\Delta$LWP up to 0.15 kg m$^{-2}$. During this period the altocumulus layer was present above the LLS at 2.5 km height. Between 07:30 UTC and 11:00 UTC $\Delta$LWP was again elevated, with values between 0.03 kg m$^{-2}$ and 0.08 kg m$^{-2}$.

The increased values of $\Delta$LWP verified the presence of liquid-water clouds as already indicated by the LLS mask. These clouds were not identified by the standard Cloudnet classification. Thus, using the standard Cloudnet classification, the radia-

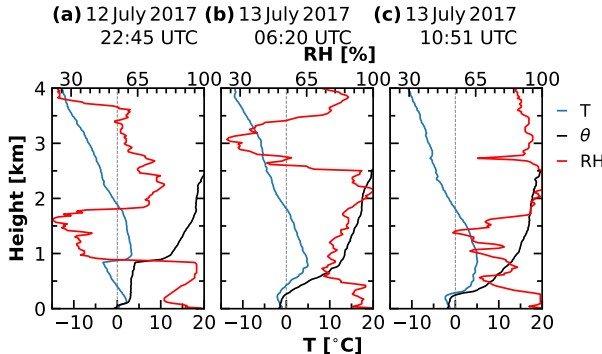

**Figure 4.** Thermodynamic profiles of temperature (blue), potential temperature (black), and relative humidity (red) up to 4 km height for three radiosonde launches relevant for the analyzed period. The start time and date are given above the respective profiles.

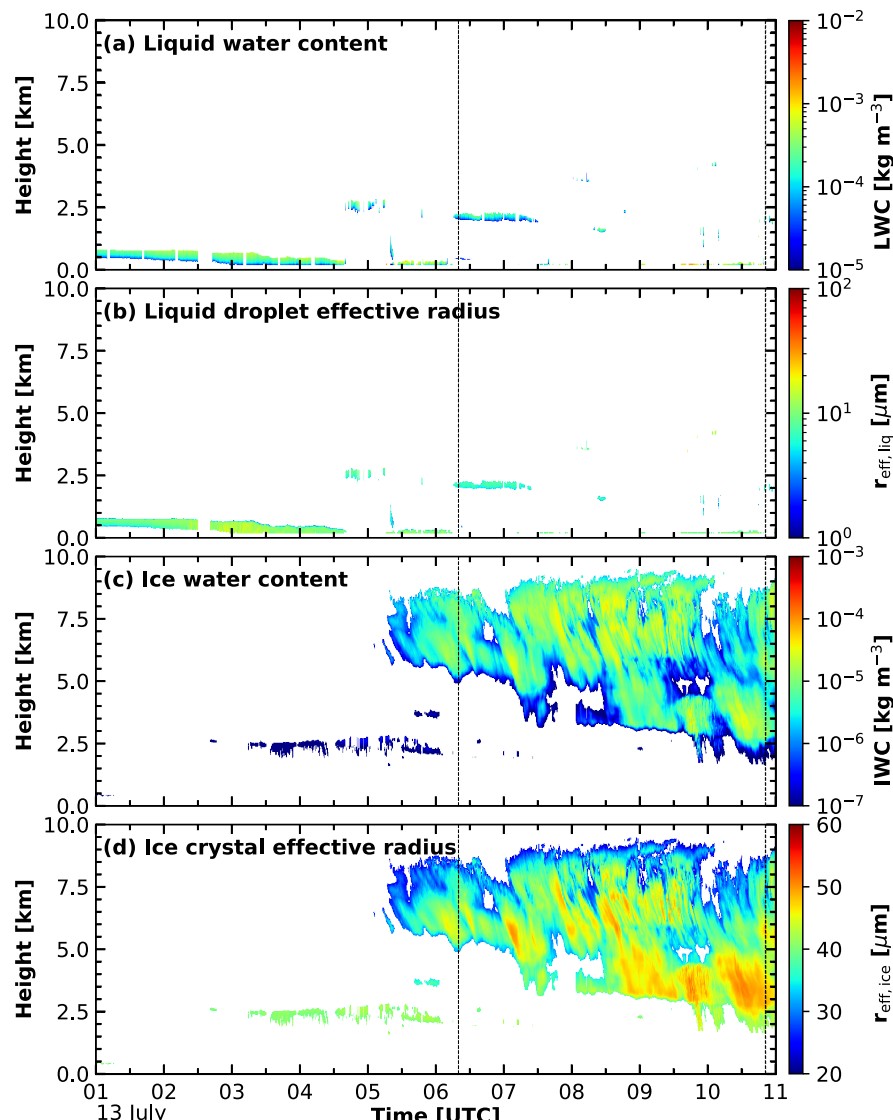

**Figure 5.** Profiles of microphysical cloud products for the same period as shown in Fig. 2 derived by Cloudnet. Panel (a) shows LWC, (b) $r_{eff,liq}$, (c) IWC, and (d) $r_{eff,ice}$. The two dashed lines mark the radiosonde launches for the profiles shown in Fig. 4 (b) and (c).

tive effect of these clouds would be calculated solely based on their ice macro and microphysical properties, leading to an underestimation of the TD and an overestimation of the SD at the surface. In the following, the effect of incorporating the
identified liquid-water clouds into the radiative transfer simulations of T-CARS is evaluated.

Three sets of radiative transfer simulations were performed to investigate the effect of the improved low-level stratus quantification. First, the Control run was conducted, applying the default Cloudnet cloud properties. In addition, a radiative simulation

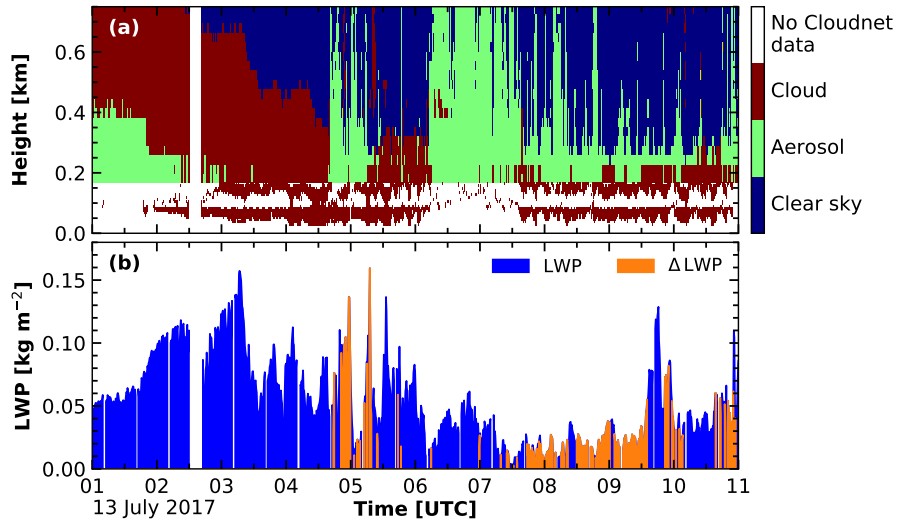

**Figure 6.** Panel (a) shows the low-level cloud mask for the same period as shown in Fig. 2 derived from the Polly[XT] near range signal (below 165 m) combined with a simplified Cloudnet classification mask (above 165 m). All clouds are shown in brown, aerosol in green and clear sky in blue. White areas denote situations where no Cloudnet data is available. In panel (b) LWP determined by the MWR HATPRO is depicted in blue. Additionally, the deviation between LWP derived by HATPRO and by the integration of LWC, $\Delta$LWP, is shown in orange.

using the improved Cloudnet input based on the approach described in Sect. 2.3 was realized. This simulation, called Scaled run, consisted of first identifying missed liquid-water clouds and then deriving their cloud droplet effective radius. Finally, also a simulation assuming a clear-sky situation was performed. The results for SD and TD at the surface are shown in Fig. 7 (a) and (c) (for clarity reasons a running mean of 5 minutes was applied). The observed SD in Fig. 7 (a) are on the first order driven by the solar zenith angle and thus follow a diurnal circle. Variations from this distribution can be caused by the presence of clouds, especially liquid-water-containing clouds. Under these cloudy conditions, the SD fluctuated from about $90\,\mathrm{W\,m^{-2}}$ at 01:00 UTC to a maximum of more than $500\,\mathrm{W\,m^{-2}}$ at around 09:10 UTC. The peaks in the SD at approximately 07:20, 09:05 and 10:00 UTC were caused by a broken cloud situation at the horizon during low solar elevation angles, identified by observations of an all-sky camera (not shown). With the appearance of LLS shortly after 04:30 UTC the simulated SD from the Control run deviated considerably from the observations. Between 04:30 and 05:30 UTC the simulated SD from the Control run reached values above $350\,\mathrm{W\,m^{-2}}$ and were similar to the clear-sky fluxes, while the observations were below $200\,\mathrm{W\,m^{-2}}$. The derived SD based on the Scaled run showed a much better agreement with the observations.

The observed TD at the surface are driven by the optical thickness of clouds and the temperature of the cloud base, which defines the respective terrestrial emission of the cloud. The stratus cloud that was present below 1 km height almost during the entire period with rather high temperatures of above $-5\,°\mathrm{C}$ caused the observed TD up to $320\,\mathrm{W\,m^{-2}}$. Deviations were observed when the LLS cloud deck was broken, at around 05:00 UTC and after 07:00 UTC (see Fig. 7 (c)). The presence of the stratocumulus at 2.5 km height with roughly the same temperature produced comparable TD. With the disappearance of the

stratocumulus and still a broken cloud situation of the LLS deck at 07:30 UTC (see Fig. 2 (e)) TD was reduced to 285 W m$^{-2}$. Same as for the SD, a clear improvement of the simulated TD using the Scaled run can be seen.

In Fig. 7 (b) and (d) the histogram of the differences in the SD and TD (simulations minus measurements) for the Scaled run (green) and the Control run (blue) are shown. Good performances of T-CARS based on the default Cloudnet classification mask applied in the Control run were derived during situations when liquid water was identified inside the clouds, e.g., until

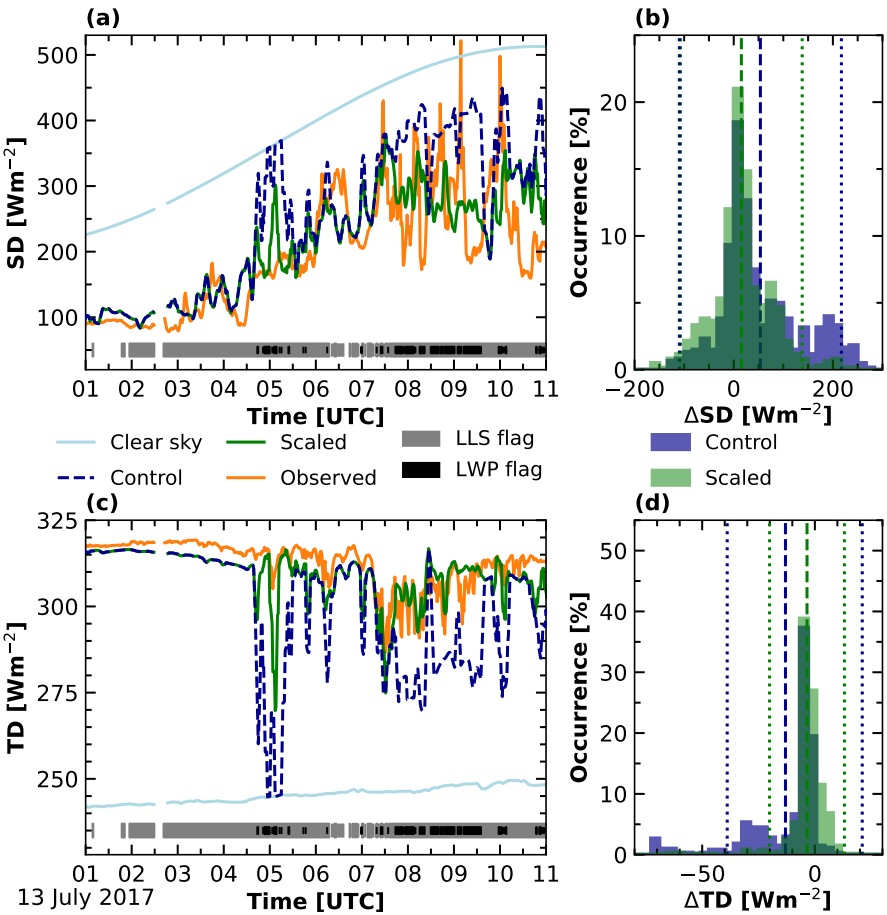

**Figure 7.** Results of the T-CARS simulations for the downward solar (SD) (a) and terrestrial radiative fluxes (TD) (c) at the surface for the same period as shown in Fig. 2. In blue, the radiative fluxes derived for the Control run and in green for the Scaled run are shown together with the respective measured values from the OCEANET pyranometer or pyrgeometer in orange. In light blue, the clear sky results from the T-CARS simulations are depicted. Additionally, the occurrence of LLS clouds is indicated by the gray LLS flag at the bottom of panels (a) and (c). The black LWP flag indicates periods with $\Delta$LWP $> 0.05$ kg m$^{-2}$. A histogram of the respective differences (simulations minus observations) is given in panels (b) and (d). The dashed lines in (b) and (d) depict the corresponding mean values and the dotted lines show the two-$\sigma$ standard deviation. The gap at 02:40 UTC is due to missing Cloudnet data.

04:30 UTC. After 04:30 UTC the differences between the measurements and the Control run were up to $+200\,\mathrm{W\,m^{-2}}$ for the SD and $-75\,\mathrm{W\,m^{-2}}$ for the TD during periods when Cloudnet failed to identify liquid and classified the clouds as pure ice. The mean downward flux difference and the respective standard deviation for the Cntrol run during the complete period analyzed was $54 \pm 82\,\mathrm{W\,m^{-2}}$ for SD and $-13 \pm 17\,\mathrm{W\,m^{-2}}$ for TD (blue dotted and dashed lines in Fig 7 (b) and (d), respectively) suggesting an underestimation of the opacity of the simulated cloud. As already indicated by the time series, applying the

approach with a more realistic LLS representation used in the Scaled run, the average differences were much smaller. In this case, the mean downward flux difference and the standard deviation for SD were $15 \pm 61\,\mathrm{W\,m^{-2}}$, and for TD $-3 \pm 8\,\mathrm{W\,m^{-2}}$ (green dashed and dotted lines in Fig 7 (b) and (d), respectively).

The resulting CRE at the surface can be derived based on the upwelling and downwelling radiative fluxes from the T-CARS simulations for all-sky and clear-sky conditions. Figure 8 (a) shows the net surface CRE for the Scaled run in green

and for the Control run in dashed blue. Diurnal variations are one of the main drivers of the atmospheric CRE during the presented period. At the beginning of the period, a slightly positive CRE was found due to the lower solar elevation angle, which turned negative at around 01:30 UTC. However, the deviations based on the adjustments in the cloud classification as already apparent in Fig. 7 (a) and (c) propagate and arise also in the net atmospheric CRE. The resulting differences in the determined CRE, Control run minus Scaled run, are shown in Fig. 8 (b). Differences between the results based on the two

approaches of up to $90\,\mathrm{W\,m^{-2}}$ were calculated, e.g., around 10:00 and 11:00 UTC. On average, the CRE based on the Control run was $-32\,\mathrm{W\,m^{-2}}$, and the CRE based on the Scaled run was $-49\,\mathrm{W\,m^{-2}}$ during the presented period. The radiative effect

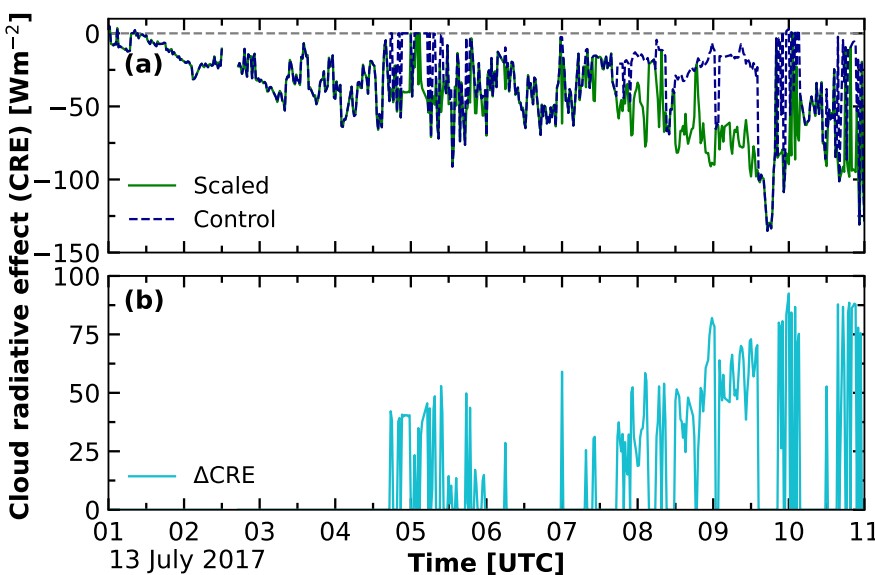

**Figure 8.** Time series of net surface CRE derived by T-CARS for the same period as shown in Fig. 2. In panel (a) the CRE derived based on the original Cloudnet classification applied in the Control run (dashed blue) and the adjusted approach used in the Scaled run (green) are depicted. In panel (b) the respective difference (Control minus Scaled) is shown.

of the LLS was determined by only considering the moments where the correction needed to be applied. During these periods throughout the presented case study, the application of the adjusted Cloudnet classification decreased the net surface CRE by $54\,\mathrm{W\,m^{-2}}$, as the cooling effect induced by the clouds in the solar range dominated over the cloud terrestrial warming in this situation.

## 3.2 Statistical analysis of the PS106 campaign

As presented in Griesche et al. (2020), LLS were observed about 25% during the PS106 observational time. In this subsection, we elaborate on the applicability of the LLS detection scheme for improving the overall radiative closure during the cruise period. We thus applied the two criteria defined in Sec. 2.3 to the full PS106 dataset, (1) the presence of an LLS layer that caused a complete attenuation of the lidar signal below the lowest Cloud radar range gate and hence prohibited the liquid cloud detection by Cloudnet and (2) a disagreement in the LWP valued derived by Cloudnet and observed by HATPRO. Overall, both criteria were fulfilled throughout 15% of the PS106 observational period. In addition, different challenges were encountered during PS106, which restricted the time where a meaningful application of the proposed method was possible, further. During approximately 9% of the PS106 observational time, shadow effects of the ship's superstructure on the radiometer measurements were determined. Also, during 18% of the time unreliable MWR measurements were identified (note that both periods may overlap). Removing these periods from the analysis, the proposed correction was applied to 11% of the observational time. In Fig. 9 the resulting relative error between the simulated and observed SD and TD radiative fluxes for the Control and the Scaled run is presented for the periods where the correction was applied. The occurrence of a relative error below 50% for the SD fluxes increased from 15.6% to 71.4% of the time, when applying the proposed correction method. For the TD fluxes, the occurrence of a relative error below 5% increased from 15.8% to 86.7% of the time. Consequently, the average relative error of the simulated SD and TD fluxes were decreased from 109% to 37% and from 18% to 2.5%, respectively.

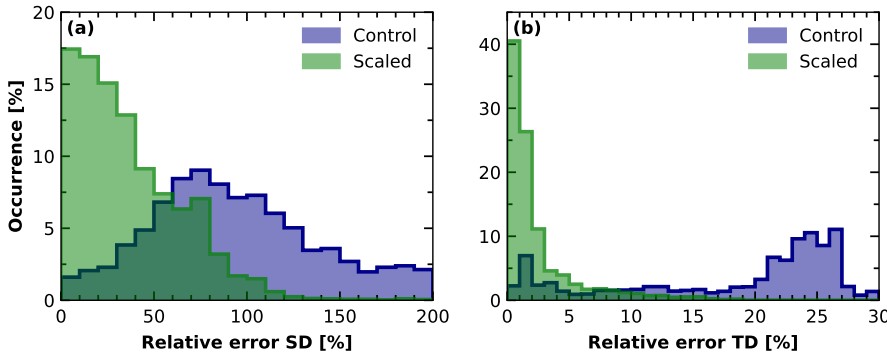

**Figure 9.** Frequency of occurrence of the relative error between the Control and the Scaled simulation and the observed radiative fluxes for the data points where the correction was applied.

## 4 Discussion and conclusion

The presented study shows the potential of obtaining more realistic radiative fluxes by using an improved liquid-water cloud detection in the input to the radiative transfer simulations. In case of a failed liquid-cloud detection, large discrepancies between the simulated and measured radiative fluxes were observed. As a reason for such a failed cloud detection, the complete attenuation of the lidar signal at height levels below the lowest detected range gate of the cloud radar (i.e., 165 meters) was identified. Clouds at such low altitudes are frequently observed in the high Arctic but are less common in lower latitudes. The instrument limitations presented in this manuscript apply also to satellite and airborne observations (Mech et al., 2019; Mioche et al., 2015; Papakonstantinou-Presvelou et al., 2022) and were an issue that was also reported in previous ground-based remote-sensing studies, as was pointed out in Sec. 1.

A key challenge encountered in this study is the complete lidar signal attenuation just above the surface and the resulting failed liquid-water detection. One potential solution would be the application of a cloud radar multipeak analysis as proposed, e.g., by Radenz et al. (2019) (peakTree), or the application of artificial neural networks (Kalesse-Los et al., 2022; Schimmel et al., 2022). These techniques, however, make use of cloud radar measurements and are therefore only applicable to higher-reaching clouds. For the low-level clouds discussed in this manuscript, with a cloud top below the lowest range gate of the cloud radar, these radar-based approaches are of limited use. Single-layer LLS, i.e., LLS without another cloud layer above 165 m, have been observed for about 5 % of the entire PS106 cruise, with daily maximum occurrence up to 40 % (Griesche et al., 2020). Together with overlaying clouds, LLS have been observed for 25 % of the cruise. The horizontal visibility sensor aboard Polarstern frequently missed these clouds as well, because the LLS base was frequently too high. Hence, the Polly$^{XT}$ near-field capabilities turned out to be crucial to detect these low-level clouds.

Besides the detection of the LLS, also the determination of the cloud microphysics for these clouds poses a challenge. Approaches for LWC and $r_{eff,liq}$, as they are, for example, implemented in Cloudnet, often rely on cloud radar reflectivity (e.g., Frisch et al., 2002; O'Connor et al., 2005). Additional approaches using lidar Raman dual-field-of-view capabilities exist meanwhile (Jimenez et al., 2020, this technique did not yet exist for the PS106 cruise). However, to apply these techniques a complete overlap between the laser pulse footprint and the receiving field-of-view is mandatory. Thus, these methods are not applicable to the discussed low-level clouds. Approaches based on active remote sensing from satellites, such as the DARDAR-CLOUD algorithm (Cazenave et al., 2019), or aircrafts, such as the observations from Ehrlich et al. (2019), suffer from ground clutter in lower altitudes and hence also struggle to observe the low-level clouds (Mioche et al., 2015; Liu et al., 2017). Passive satellite products, such as the SYN1deg from Clouds and the Earth's Radiant Energy System (CERES) often have significant errors in their retrieved parameters and the vertical structure of multilayer cloud conditions (Rutan et al., 2015; Minnis et al., 2019; Yost et al., 2021). Hence, here the liquid-water cloud microphysical properties were derived by an effective analysis of the LWP for surface-coupled, single-layer liquid clouds and a statistical analysis of $r_{eff,liq}$ determined during the PS106 cruise. The calculated values of the radiative fluxes during the presented case study suggest that by applying the adjusted method radiative closure is achieved for the analyzed period. The mean flux differences (SD: 15 W m$^{-2}$, TD: $-3$ W m$^{-2}$) were below the instrumental uncertainties (i.e., $\pm20$ W m$^{-2}$ pyranometer (SD) and $\pm10$ W m$^{-2}$ for pyrgeometer (TD), (Lanconelli

et al., 2011)). Due to the short period analyzed, the presented case can not be seen as representative of the Arctic other than for the specific time and location. The results still fit into previous studies on the broader picture of the CRE in the Arctic, for example, reported by Shupe et al. (2015), Ebell et al. (2020) and Barrientos-Velasco et al. (2022). A direct comparison between the simulated and the observed CRE was not possible during PS106, as only the downward radiative fluxes were measured.

The presented findings demonstrate that a detailed characterization of the low-level clouds can significantly improve the quality of radiative transfer simulations. In the standard configuration of processing schemes for cloud microphysical products, such as Cloudnet, these low-level clouds are often underrepresented. Unless otherwise considered, this lack of low-level clouds eventually leads to large differences in the calculated CRE. A difference in the derived surface CRE of up to $90 \, \mathrm{W \, m^{-2}}$ was calculated in the presented case study when these low-level clouds were considered in T-CARS input compared to the

application of the standard Cloudnet products. Comparing the Control and the Scaled simulation, a positive contribution of the low-level liquid-containing cloud to the surface net CRE of $54 \, \mathrm{W \, m^{-2}}$ was found. The result underlines the findings of previous radiative studies, which have shown that low-level clouds are of great importance for the Arctic radiation budget. An accurate representation of these clouds in radiative simulations is hence a prerequisite for the understanding of a piece of the puzzle of Arctic amplification. The presented case also highlights the importance of accurate radiative flux measurements.

The approach to consider the missed liquid-containing clouds in the radiative transfer simulations by constant values of the droplet effective radius already reduced the mean difference between the modeled and measured radiative fluxes below the measurement uncertainty.

The introduced criteria for the correction of the LLS, identified LLS in addition with a disagreement of the LWP from Cloudnet and HATPRO, were observed in total during 15% of the observational time. However, during PS106 different limitations

were encountered, such as unreliable radiative flux measurements due to shadowing effects caused by the ship's superstructure, precipitation on the radiometers, or very low sun elevation angles. Possible shadowing effects of the ship's superstructure during the PS106 campaign were identified during 9% and unreliable LWP values during 18% of the time (both may have occurred at the same time). Removing these periods from the analysis, the proposed correction was applied to 11% of the data. For these data, the relative error was then reduced on average from 109% to 37% for SD and from 18% to 2.5% for TD fluxes.

To challenge the remaining deviations of the observed and simulated radiative fluxes, future studies should, for example, also consider spatial cloud homogeneity. Especially, when analyzing mobile observations in the Arctic, where the complexity among the interaction of variable low sun angles, multiple reflections due to cloud inhomogeneities, and variable surface conditions (i.e., open ocean, marginal ice zone, ice surfaces) increases. Therefore, extending the analysis to improve the understanding of 3-D radiative effects is also recommended. The difference between 3-D and 1-D simulations has been assessed by,

e.g., Barker et al. (2012). In their study, the authors compared the results of the two different simulations based on satellite cloud observations. Incorporating 3-D effects into radiative transfer simulations, they found differences for the solar surface radiative fluxes of up to $30 \, \mathrm{W \, m^{-2}}$, compared to simulations that only apply 1-D effects. Additionally, ground-based remote sensing is frequently performed in a vertical or almost vertical direction. Hence, the divergence of the viewing angle of ground-based remote-sensing instruments and the solar zenith angle can cause differences in the cloud properties applied in the radiative

transfer simulations and the cloud situation influencing the measured radiative fluxes. Accordingly, 3-D scanning remote sens-

ing with lidar and radar can be expected to provide improvements in both, the detection of cloud inhomogeneities as well as in the detection of LLS.

Further investigation is needed to tackle the above-mentioned remaining deviations and to further investigate the radiative transfer simulations based on the derived cloud properties for different meteorological scenarios. In order to do so, the presented approach will be applied to observations from the year-long MOSAiC expedition (Engelmann et al., 2021; Shupe et al., 2022). MOSAiC was an ice-breaker-based expedition, conducted from fall 2019 until fall 2020 in the high Arctic, and has achieved unprecedented detailed observations of the Arctic system. The comprehensive extent of the MOSAiC expedition offers the possibility to minimize the limitations encountered during PS106. During the MOSAiC expedition, sufficient amount of good-quality data was collected to cover different meteorological scenarios along a full annual cycle. Low-sun elevation angles may only play a role during the summer half of the expedition period. Additionally, there were several broadband radiometers installed on board Polarstern and in the vicinity of the icebreaker on the ice floe (e.g., Cox et al., 2023; Riihimaki, 2023), which will allow a complete comparison without the interference of the ship's superstructure. The combination of upward and downward-directed radiative flux measurements performed during MOSAiC will enable us a direct comparison of the CRE. These measurements were not only performed at the surface but also, for example, with tethered balloon systems (Lonardi et al., 2022). Also, to contrast the effects of low-level clouds on the radiative budget in the Arctic and Antarctic, we will apply the presented method to data from the COALA (Continuous Observations of Aerosol-cLoud interaction in Antarctica) project. In the framework of COALA, the OCEANET-Atmosphere suite is deployed for one year at the Germany Antarctic station Neumayer III (70.65° S, 8.25° E, height above sea level: 43 m, WMO code: 89002), which enables us a similar analysis as presented here.

In conclusion, the following key statements from our study remain. Through our literature review, we have demonstrated that quantifying the abundance and determining the properties of the lowest-level Arctic clouds, ranging from just above the surface to about 150 m still poses challenges on remote-sensing approaches from the ground, aircraft, and satellite. The main reason for this gap is, that current remote-sensing techniques, or their implementation, struggle to disentangle the effects of the lowest clouds from higher-reaching ones. Hence, current retrievals have difficulties to provide a detailed characterization of these low-level clouds. When such clouds are missed from the analysis, considerable biases in the determination of CRE occur. During the summer months over the marginal sea ice zone, such clouds were, e.g., found to be present during 25% of the observation time. LWP-thresholding and incorporation of optimized near-range lidar data are promising approaches to enable a thorough representation of LLS. Further improvement can be expected by using enhanced observational capabilities such as scanning radar or lidar, as well as by a transition from 1-D to 3-D radiative transfer modeling for capturing cloud inhomogeneities.

*Data availability.* The Cloudnet data are published via the Pangaea data archive and available under Griesche et al. (2019, 2020a, b, c, d, e, f, g). The radiative transfer simulations are published via Zenodo and available under Barrientos-Velasco (2023) for the case study and under Griesche (2023) for the complete PS106 campaign.

*Author contributions.* HG and CBV developed the study concept. HG led the data analysis and interpretation, and performed the Cloudnet processing. CBV performed the radiative transfer simulations. All authors actively contributed to the analysis and discussion. HG wrote the manuscript with input from all co-authors.

*Competing interests.* The contact author has declared that none of the authors has any competing interests.

*Acknowledgements.* The authors gratefully acknowledge the funding by the Deutsche Forschungsgemeinschaft (DFG, German Research Foundation) – Project Number 268020496 – TRR 172, within the Transregional Collaborative Research Center "ArctiC Amplification: Climate Relevant Atmospheric and SurfaCe Processes, and Feedback Mechanisms (AC)3". We also acknowledge the funding from Bundesministerium für Bildung und Forschung for the project Combining MOSAiC and Satellite Observations for Radiative Closure and Climate Implications (MOSARiCs) (Project Number 03F0890A). The authors also acknowledge support through ACTRIS-2 under grant agreement no. 654109 from the European Union's Horizon 2020 Research and Innovation Programme.

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
