# Peer review of "Low-level Arctic clouds: A blind zone in our knowledge of the radiation budget"

_EGUsphere, 2023_

## Author Response (AR1)

**Reply to Review #1**

Dear Reviewer,

We thank you for carefully reading through the document and providing us with comments to improve the manuscript! We have prepared a consolidated response to two of the major comments of both anonymous reviews as their concerns were aligned: the limited period analyzed in the study. The line numbers refer to the diff-version of the manuscript, which is attached to this reply letter.

The analysis of further cases or the entire PS106 data would likely have strengthened our analysis. In the course of this study, we have analyzed several cases from the PS106 campaign with low-level stratus (LLS) present, in order to test our approach for different situations and to constrain the effect of LLS on the surface radiation budget. We have selected this case study out of the manifold scenarios we have observed during the PS106 expedition due to its ideal conditions for our purpose. In the presented case, a homogeneous and continuous LLS layer was present, while no precipitation at the surface or any interference of the ships' superstructure with the measurements was observed. This particular situation was just right to determine the influence of the LLS on the radiation surface budget.

During the other analyzed cases, however, we encountered additional challenges besides the presence of LLS, which interfered with assessing the LLS effect. Therefore, these cases could not serve the purpose of this study, and we thus focused on the one presented. The challenges encountered in the other cases were as follows:

- Shadow effects from the ships' superstructure and crane operations interfered with the radiative flux measurements.

- Very low sun elevation angles and spatial cloud inhomogeneities caused discrepancies between the cloud situation described by the Cloudnet products and the measured incoming solar radiation (3-d effects).

- Unreliable radiometer measurements were observed due to precipitation and yet to be determined artifacts affecting the surface radiation flux measurements.

The handling of these challenges and the assessment of their influence on the cloud radiative effect are subjects of other projects and were not within the scope of this study.

To this end, we will apply the proposed method to further campaigns. A focus will be the usage of the data from the year-long Multidisciplinary drifting Observatory for the Study of Arctic Climate (MOSAiC) expedition, which was conducted from fall 2019 until fall 2020 in the high Arctic. The MOSAiC expedition achieved an unprecedented level of detailed observations of the Arctic system. The extensive duration of the expedition and the comprehensive suite of observations will allow us to better understand the effects of the encountered challenges. With the MOSAiC dataset, a direct comparison of the simulated and observed cloud radiative effect is possible, as both the downward and upward-directed

radiative fluxes were measured. These measurements were not only conducted at the surface but also, for example, with tethered balloon systems, enabling the assessment of the cloud radiative effect at higher altitudes. The utilization of various pyranometers and pyrgeometers during the MOSAiC expedition additionally offers the opportunity to minimize instrument limitations.

In addition, the same instrumentation suite used during the PS106 campaign is currently deployed at the German Antarctic Station Neumayer III for one year as part of the Continuous Observations of Aerosol-cLoud interaction in Antarctica (COALA) project. The derived dataset will be utilized in a similar manner as presented in this study to contrast the radiative effects of low-level clouds in the Arctic and Antarctic.

We have included a paragraph in the revised version of this manuscript (lines 350-365) that describes the encountered challenges as well as the necessity for further studies in this regard, and, as requested by Reviewer #2, outlines the plan to apply the method to the MOSAiC data.

Below, we provide a step by step reply to the minor comments. The reviewer comments are given in black and our answers in blue. The line references refer to the diff version of the revised manuscript.

lines 119-: Can you remind the reader how exactly LWC and reff is retrieved in Cloudnet?

- We added a section on the retrievals of LWC, liquid droplet effective radius, IWC, and ice crystal effective radius (lines: 135-140).

lines 127-: "and the values for the HATPRO LWP and $LWP_{LWC}$ differ."

In this case, $LWP_{LWC}$ is simply zero, since Cloudnet does not provide an LWC profile, right? Maybe this could be clarified in the text.

- Indeed, the LWC simply is zero in these cases. We have clarified this (line: 176).

line 144: There is no section 3.2. So subsection 3.1 is not needed. However, a section 3.2 could be added which includes an analysis of the whole PS106 time period, for example.

- Subsection 3.1 is removed. See reply to major concerns to the proposed idea of an analysis of the whole PS106 campaign.

line 178: "In green,…" This sentence can be deleted.

- This sentence is deleted.

lines 190-191: "Smaller deviations […] at around 5 UTC…"

I would not call the differences around 5 UTC small.

- We removed the word 'small'.

**Reply to Review #2:**

Dear Reviewer,

We appreciate your careful review of the document and your valuable comments, which have helped improve the manuscript. In response to the major comments raised by both anonymous reviewers, we have prepared a consolidated response as their concerns were aligned, specifically regarding the limited period analyzed in the study. Our reply to this specific point of Review #2 is thus the same as the one to the same point of Review #1, and is therefore just repeated below. Anyway, replies which are specific to Review #2 will of course follow below. The line numbers refer to the diff-version of the manuscript, which is attached to this reply letter.

The analysis of further cases or the entire PS106 data would likely have strengthened our analysis. In the course of this study, we have analyzed several cases from the PS106 campaign with low-level stratus (LLS) present, in order to test our approach for different situations and to constrain the effect of LLS on the surface radiation budget. We have selected this case study out of the manifold scenarios we have observed during the PS106 expedition due to its ideal conditions for our purpose. In the presented case, a homogeneous and continuous LLS layer was present, while no precipitation at the surface or any interference of the ships' superstructure with the measurements was observed. This particular situation was just right to determine the influence of the LLS on the radiation surface budget.

During the other analyzed cases, however, we encountered additional challenges besides the presence of LLS, which interfered with assessing the LLS effect. Therefore, these cases could not serve the purpose of this study, and we thus focused on the one presented. The challenges encountered in the other cases were as follows:

- Shadow effects from the ships' superstructure and crane operations interfered with the radiative flux measurements.

- Very low sun elevation angles and spatial cloud inhomogeneities caused discrepancies between the cloud situation described by the Cloudnet products and the measured incoming solar radiation (3-d effects).

- Unreliable radiometer measurements were observed due to precipitation and yet to be determined artifacts affecting the surface radiation flux measurements.

The handling of these challenges and the assessment of their influence on the cloud radiative effect are subjects of other projects and were not within the scope of this study.

To this end, we will apply the proposed method to further campaigns. A focus will be the usage of the data from the year-long Multidisciplinary drifting Observatory for the Study of Arctic Climate (MOSAiC) expedition, which was conducted from fall 2019 until fall 2020 in the high Arctic. The MOSAiC expedition achieved an unprecedented level of detailed observations of the Arctic system. The extensive duration of the expedition and the

comprehensive suite of observations will allow us to better understand the effects of the encountered challenges. With the MOSAiC dataset, a direct comparison of the simulated and observed cloud radiative effect is possible, as both the downward and upward-directed radiative fluxes were measured. These measurements were not only conducted at the surface but also, for example, with tethered balloon systems, enabling the assessment of the cloud radiative effect at higher altitudes. The utilization of various pyranometers and pyrgeometers during the MOSAiC expedition additionally offers the opportunity to minimize instrument limitations.

In addition, the same instrumentation suite used during the PS106 campaign is currently deployed at the German Antarctic Station Neumayer III for one year as part of the Continuous Observations of Aerosol-cLoud interaction in Antarctica (COALA) project. The derived dataset will be utilized in a similar manner as presented in this study to contrast the radiative effects of low-level clouds in the Arctic and Antarctic.

The reference to these plans were indeed missing in the manuscript, as pointed out in your second major concern. We have therefore included a paragraph in the revised version of this manuscript (lines 350-365 [in the diff-version]) that describes the encountered challenges as well as the necessity for further studies in this regard, and that outlines the ideas to apply the presented method to the MOSAiC and COALA data.

Below, we provide a step by step reply to the specific comments. The reviewer comments are given in black and our answers in blue. The line references refer to the diff version of the revised manuscript.

The study concluded that representation of these low-level clouds in the radiative transfer simulations let to errors in the cloud radiative effect of 43Wm−2, leading to an improved representation of surface radiation budget. All conclusions are drawn from one case study. To make conclusions more robust, it is necessary to perform calculations in different time and location.

Is proposed adjusted classification scheme applicable to other conditions or field campaigns?

- Yes, it is planned to apply the proposed method to the observations of the MOSAiC expedition and the COALA project. We clarified this in the manuscript, see lines 355-365.

In addition to 13 July 2017 case, do you have other cases to quantify the impact of adjusted Cloudnet classification scheme on surface radiative fluxes? How representative is this case to the summertime cloud conditions in the Arctic?

- We have also studied other low-level stratus cases from the PS106 campaign. However, as pointed out in the reply to the major concerns, external effects prevented a meaningful analysis of these cases.

Line 34-48: A more recent study also performed radiative transfer simulations using measurements from MOSAiC field campaign to quantify the uncertainties in Arctic surface radiation budget.

Reference:

Huang, Y., P.C. Taylor, F.G. Rose, D.A. Rutan, M.D. Shupe, and M.A. Webster. (2022): Towards a more realistic representation of surface albedo in NASA CERES satellite products: a comparison with the MOSAiC field campaign. Elementa: Science of the Anthropocene, 10 (1): 00013.

- We have included this reference in the introduction (lines: 63-68).

Section 2.3: Can you mark the location of the observations taken on 13 July 2017 in a map?

- A map was included (Fig. 1) and the location on July 13 was marked.

Line 120: Please correct this sentence: "The cloud radar measurements, however, are limited to altitudes above 165m above the ground."

- This sentence was changed to: "The lowest range gate of the cloud radar, however, is located 165 m above the ground." (lines: 168-169)

Line 206-216: Is CRE calculation improved with adjusted Cloudnet classification scheme? Can you compare simulated CRE with the observations?

- Unfortunately, no upward directed radiative fluxes were measured during PS106. Therefore, no direct assessment of the simulated CRE is possible. This, however, will be done for the MOSAiC observations.

Line 229: "no qualitative assessment of the microphysical properties of these low-level clouds has been conducted." Do you want to say "quantitative" here?

- We have removed this sentence in response to the third review, where it was pointed out that satellite-based studies already gave some assessment of low-level clouds.

Section 4 Discussion and Conclusions: It would be better to re-organize this section. The challenges previously identified in field campaigns and studies would be better suited for the motivation section. The future direction of the study should follow the conclusions drawn from the current research.

- Thanks for the remark. Indeed, parts of the discussion were better suited for the introduction and was moved there (lines: 75-93).

**Reply to Review/comment #3 by Luca Lelli**

Dear Luca,

We appreciate your comment on our manuscript and for bringing to our attention certain statements where we may have missed the intended point. The references you provided were indeed helpful in clarifying our message.

We acknowledge the significant advantage of satellite missions in their ability to cover large spatial scales, as opposed to the point measurements typically obtained from ground-based instruments. However, we agree that some of our statements may have been overly emphasized. The figure you provided in your comment highlights the promising advancements in low-level cloud detection achieved by satellites. In response, we have expanded the paragraph discussing various satellite-based approaches, their strengths, and weaknesses (lines: 35-51 [all line specifications refer to the diff-version]).

The statements which were referred to in the comment:

'In both the introduction (lines 54-56: "Yet ... missing.") and the conclusion (lines 298-301: "In conclusions ... clouds") it is stated that clouds, whose altitude is very low (~150 m, if I am not mistaken), cannot be characterized by means of satellite retrievals.'

were revised and read now as:

*An Arctic-wide quantification of these low-level clouds and the disentangling of their radiative effects from those from higher clouds is still difficult.* (lines: 94-95)

and

*In conclusion, the following key statements from our study remain. Through our literature review, we have demonstrated that quantifying the abundance and determining the properties of the lowest-level Arctic clouds, ranging from just above the surface to about 150 m still poses challenges on remote-sensing approaches from the ground, aircraft, and satellite. The main reason for this gap is, that current remote-sensing techniques, or their implementation, struggle to disentangle the effects of the lowest clouds from higher-reaching ones and hence provide a detailed characterization of these clouds.* (lines: 366-371)

We hope that these statements, along with our review of the latest studies, now offer a more realistic representation of the current state-of-the-art regarding the satellite (as well as ground-based and aircraft) capabilities to derive cloud properties in the Arctic.

---

## Author Response (AR2)

**Reply #2**

Dear Reviewer,

We thank you for the comments on the revised version of the manuscript. Choosing case studies is probably always a way of cherry picking. However, we fully agree that we want to show the comprehensive benefit of the proposed method, despite the challenges we faced while applying it to the whole PS106 campaign data. Hence, we ran the radiative transfer simulation with the improved input for the whole campaign period. We have extended the manuscript with an analysis of how often it was possible to apply the proposed method and on the frequency of the challenges. We show how the proposed method has increased the quality of the radiative transfer simulations during the periods where the correction could be applied.

In the course of applying the requested revisions we have noticed an inconsistency in the calculation of the radiative effect of the LLS during the presented case study. In the first versions of the manuscript, we have reported the deviation between the simulation and the observation for the whole day of the case study, not, as it was supposed to be, for the presented period only. Based on these numbers, we have then calculated the radiative effect of the LLS as difference of the two simulation runs. In the updated manuscript we now report the deviation between the simulation and the observation for the period of the case study only. Finally, to derive the effect of the LLS on the radiative budget, we have averaged the differences between the two simulations only for those moments where the criteria as proposed in the manuscript in Section 2.3 (LLS present in addition with different LWP from Cloudnet and HATPRO) were fulfilled. The text in the manuscript was revised accordingly (lines 261-267 in the diff version of the manuscript).

Here our detailed answers to your comments (comments from the Report #2 are given in black, our replies in blue). We have updated the manuscript accordingly and provide line numbers to the diff version of the revised manuscript.

Comment 1
Can you provide some statistics in the manuscript about how often low-level stratus cloud situations occur, for how many cases the cloud properties retrieval method can be applied, for how many cases not and for which reasons (% values)

The frequency of LLS occurrence was presented in Griesche et al. (2020) and was determined as 25% of the observation time. The criteria for the LLS correction presented in this manuscript (LLS occurrence + LWP disagreement) were fulfilled during 15% of the time.

The frequency of occurrence of the following challenges was determined:

- Ship superstructure caused shadowing of the radiometers: 10% of the time
- No reliable LWP: 18% of the time

(Note that both challenges may have been occurred during the same time.)

After removing these periods, the criteria for the LLS correction (LLS occurrence + LWP disagreement) were fulfilled during 11% of the observational time. For these periods we have applied the proposed correction of the LLS and did run the radiative transfer simulations.

Based on these data, we have calculated the relative error for both the standard and the scaled T-CARS input for the solar (SD) and terrestrial (TD) downward radiative fluxes after removing periods where shadow effects and unreliable LWP were detected. The frequency of occurrence of the respective errors for those periods where the correction was applied is shown in Figure 1. The occurrence of a relative error below 50% for the SD fluxes increased from 15.6% to 71.4% of the time, when applying the proposed correction method. For the TD fluxes, the occurrence of a relative error below 5% increased from 15.8% to 86.7% of the time.

[Figure]

Figure 1: Frequency of occurrence of the relative error between the Control and the Scaled simulation and the observed broadband downward radiative fluxes for the data points where the correction was applied during PS106.

The manuscript was updated accordingly. We added a new Subsection 3.2 (lines 268 - 283) and revised the Abstract (lines 13 - 15) and the Discussion and conclusion (lines 335 - 342) accordingly.

Comment 2
Why are you sure that this method will better work for the MOSAiC measurements? In many situations (very low sun angle, broken clouds,…) you will probably have the same issues. Can you already provide corresponding numbers as mentioned in comment 1 but for the MOSAiC data set to give more confidence in the applicability for upcoming studies as mentioned in the outlook?

During the year-long MOSAiC expedition, we were able to collect a sufficient amount of good-quality data to cover different meteorological scenarios during a complete annual cycle. Low-sun elevation angles may only play a role during the summer half of the expedition period. Additionally, there were installed several broadband radiometers on board Polarstern and in the vicinity of the icebreaker on the ice floe, which will allow a complete comparison without the interference of the ship's superstructure. We revised the paragraph on the MOSAiC plans in the Discussion and conclusion accordingly (lines 360 - 369).

Comment 3

You should consider the application of the improved cloud retrieval method independently of the fact that the 1D radiative transfer simulations can be applied or not. You can still apply the cloud retrieval to all PS106 cases where possible and provide some overall statistics of Arctic LLS properties. The radiative closure study/analysis of cloud radiative effect might not be possible for all these cases due to for example 3D effects but this is another issue and might be solved in future by using a 3D radiative transfer code. Having an improved cloud microphysical data set of LLS is already quite an achievement. But here you need to clearly demonstrate the applicability. What are the limitations and how could you improve this in future (related to comment 1)?

The LLS retrieval was already applied to the whole PS106 campaign, and the results were presented by Griesche et al. (2020). We apologize that this was not made clear before. The manuscript has been updated accordingly (lines 114 and 271). Related to this comment, please see also the answer to Comment 1 for more details. In the future, we aim to automatize the method and embed it into the coding routines of the Cloudnet algorithm for the MOSAiC expedition. In this way, the Cloudnet microphysical products tailored for the Arctic region would already be corrected for the LLS occurrence in the Arctic and could, for example, be applied in comparable radiative closure studies or other cloud related studies. We extended the discussion on the MOSAiC data to inform the reader about these plans (lines 360 - 369).